# Symbiotic and Asymmetric Causality of the Soil Tillage System and Biochar Application on Soil Carbon Sequestration and Crop Production

**Amare Assefa Bogale [1,2,\*], Anteneh Agezew Melash [3] and Attila Percze [1]**

[1] Institute of Crop Production, Hungarian University of Agriculture and Life Sciences, 2100 Gödöllő, Hungary; percze.attila@uni-mate.hu

[2] Department of Horticulture, College of Agriculture and Natural Resource, Mekdela Amba University, Tulu Awulia P.O. Box 32, Ethiopia

[3] Faculty of Agricultural and Food Sciences and Environmental Management, Institute of Crop Sciences, University of Debrecen, Boszorm'enyi Str. 138, 4032 Debrecen, Hungary

\* Correspondence: bogale.amare.assefa@phd.uni-mate.hu

**Abstract:** Agriculture faces a significant challenge in maintaining crop production to meet the calorie demand of the ever-growing population because of limited arable land and climate change. This enforces a search for alternative multifarious agricultural-based solutions to meet the calorie demand. In search of alternatives, agricultural soil management has been highlighted and is expected to contribute to climate change mitigation through soil carbon sequestration and reduce greenhouse gas emissions through effective agricultural management practices. The addition of biochar to the soil significantly improves the soil nitrogen status, soil organic carbon, and phosphorus, with greater effects under the different tillage systems. This symbiosis association could further change the bacterial structure in the deeper soil layer which thus would be important to enhancing productivity, particularly in vertisols. Biochar also has an environmental risk and negative consequences. Heavy metals could be present in the final food products if we use contaminated raw materials to prepare biochar. However, there is a need to investigate biochar application under different climatic conditions, seasons, soil tillage systems, and crop types. These indicate that the positive effect of proper biochar fertilization on the physiology, yield formation, nutrient uptake, and soil health indicators substantiate the need to include biochar in the form of nutrients in the crop production sector, especially in light of the changing climate and soil tillage systems.

**Keywords:** biochar; carbon sequestration; conservation tillage; conventional tillage; greenhouse gases

## 1. Introduction

In agriculture, crop production is a prominent sector intended to ensure food security, enhance food self-sufficiency, and supply input for different industries worldwide. However, there is a need to enhance the productivity of the crop production sector to satisfy the calorie demand of the world's ever-increasing population. Hence, to meet this demand, the producers should keep in mind the soil health of where the crops are grown and the environment in which the living things survive, since soil management is an important aspect affecting the functionality of the soil [1–3]. However, these aims have been hampered due to improper agronomic management practices and environmental degradation. It has been observed that the degradation of soil, due to the loss of soil carbon (C) and nitrogen (N) pools, is decreasing crop productivity and intimidating food security [1]. Tillage is one of the critical components of agricultural systems and practices, commonly working internationally in croplands to reduce climatic and soil restrictions, even though sustaining several ecosystem services is an issue [4]. This practice can affect several soil-mediated processes, such as soil carbon sequestration (SCS) or depletion, water pollution, and greenhouse gas ($CO_2$, $CH_4$, and $N_2O$) emissions [4]. Maximum and intensive tillage without

proper residues and crop nutrient management practices is among the reasons for the loss of soil C and N pools and the decreased productivity and quality of the crops [1].

In addition to tillage's effect on the soil nutrient status, it can also further influence $N_2O$ emissions by affecting the nitrification and denitrification processes, influence $CH_4$ emissions by affecting methanotrophs [5], and impact the quantity and composition of the soil organic matter [6]. Increments of $N_2O$ and $CH_4$ are greenhouse gases with a global warming potential of approximately 300 and 30 times that of $CO_2$ [7,8]. However, both $CH_4$ and $N_2O$ emissions could be counterbalanced through the positive benefits gained by soil carbon sequestration practices. Hence, enhancing soil organic carbon through proper implementation of agronomic crop management practices is very important to increasing organic matter inputs into the soil, reducing the decomposition of soil organic matter, and oxidizing soil organic carbon; each of these, or a combination of them, act as an alternative agronomic practice for a sustainable agriculture production system designed to enhance overall crop productivity [9,10].

A number of agronomic practices, such as biochar application, have been previously proposed to be aimed at mitigating greenhouse gas emission and simultaneously improving the yield and quality of crops. Biochar is fine-grained charcoal made up of a stable carbon-based material and applied to soils to realize the net carbon sequestration, which is encouraged principally for balancing the effect of climate change and maintaining the soil fertility status to improve the yield of the crops [1]. It can be prepared from organic materials, including paper mill sludge, crop and forestry residues, and poultry waste. The adoption of biochar is gaining growing attention as a sustainable technique that helps improve soil fertility in weathered and degraded soils [11]. It consists of a stable C compound produced via pyrolysis and gasification [12,13]. Biochar mainly subsidizes the stable SOC pool, although fresh crop residues add to the more bio-accessible fraction of the soil C reserve. This is why the decomposition of biochar proceeds naturally at a rate that is 10 to 100 times slower than uncharred biomass [12,14]. Therefore, our hypothesis is that the combination of a soil tillage system and biochar application can have a symbiotic effect on soil carbon sequestration and crop production. Specifically, the effects of each practice may be enhanced by the other, resulting in greater benefits than if they were used independently. For example, the use of biochar can mitigate the negative effects of tillage on soil structure, while the use of tillage can increase the availability of nutrients for plant growth. Hence, the objective of the review is to illustrate the independent and symbiotic effect of soil tillage systems and biochar on crop production, carbon sequestration, and other soil properties.

## 2. Materials and Methods

Several techniques were implemented to assure a high-quality literature review. The first step was selecting the topic or the title and placing the subtitles under the main topic. This review paper's subtitles include 'the effect of conventional tillage systems on crop production', 'the effect of conservation tillage systems on crop production', 'carbon sequestration as influenced by those tillage systems and the role of biochar for crop production', 'the effect of biochar in the environment', and 'soil carbon sequestration'. Then, based on the selected titles, the second step was collecting different scientific articles from different sources, such as Google® Scholar, PubMed, ResearchGate, Web of Science, and Scopus. Thirdly, the selected articles were organized based on their relevancy and latest publication year. Following thorough reading of the selected articles, a review was composed based on the ideas that were relevant to the topics. Only English-language articles were taken into consideration for this review.

## 3. Overview of Soil Tillage System and Its Role in Agriculture

Identification of an environmentally friendly and crop-yield-sustaining tillage strategy is required to remedy the growing concern for food security through enhanced soil management practices [15]. The mechanical alteration of soil for crop production, also known as tillage, has dramatically changed soil properties, including soil temperature,

infiltration, and evapotranspiration. This intentional soil disturbance during tillage causes an impact on the ecosystem and the growing of crops [15,16]. It is an important component of agricultural technology [17–20] because it alters the primary soil layer's root zone, the soil is nourished with mineral or organic fertilizer and plant leftovers, and a good seedbed is made [20,21]. Soil tillage is a crop production factor contributing up to 20% [22,23] and affecting the sustainable use of soil resources through its influence on the physical and chemical properties of soil [24]. This indicates that tillage is a fundamental crop production practice that emphasizes forming a good seedbed for germination and subsequent plant growth; there are changes in the soil bulk density and resistance, improving soil aeration and providing ideal conditions for plant life. Hence, based on this information, our intentions are concerned with conservation tillage and the conventional tillage systems as follows.

### 3.1. Effect of Conservation Tillage on Soil Properties

Conservation tillage can be defined as a crop management strategy that leaves at least 30% of the crop residue on the soil's surface after planting to prevent soil erosion by water [25]. Conservation tillage systems are a part of the practice of managing agricultural leftovers on the soil's surface with minimum or no tillage. The techniques are also referred to as direct drill, stubble mulching, eco-fallow, restricted tillage, reduced tillage, minimum tillage, and no tillage. These approaches to plant residue management have three main objectives: lowering energy use, saving soil and water, and often leaving enough plant residue on the soil's surface to minimize wind and water erosion [26]. It is worth mentioning that conservation tillage also has the potential to reduce the negative effects of soil disturbance and nutrient loss, which arise due to erosion, increased rainfall infiltration, reduced subsurface compaction, and maximized soil organic carbon (SOC) accumulation, with this mitigation positively affecting many soils' physical and chemical properties [27].

The magnitude and the level of influence of conservation tillage practices on soil nutrient status have been observed as divergent, according to a given set of environmental conditions. In an environment where wind erosion is an issue, conservation tillage refers to any strategy that keeps at least 1000 pounds per acre of crop residue from small grains on the surface during the crucial erosion period [28]. It has been observed that conservational tillage practices can also attain the highest organic matter accumulation, result in the maximum root mass density (0–15 cm soil depth), and improve the physical and chemical properties of the soil. However, bulk densities were reduced due to the tillage practices, having the highest reduction of these properties and the highest increase of porosity and field capacity, particularly at zero tillage. Zero tillage with 20% residue retention, is suitable for soil health and attaining the optimum yield [23]. According to [1], conservation tillage with proper nutrient management and crop residues improves rice yield by 51.1–52.2%. This indicates that conservation tillage contributes significantly to improving grain yield compared to conventional tillage. On the other hand, the no-tillage system can also provide several benefits, including enabling the growers to manage more significant areas of cropland with less total input [29], reducing the soil erosion caused by wind and water, and enabling the accretion of soil organic carbon (SOC) and an improved soil structure [30]. Although the no-tillage system provides all of the aforementioned benefits, there are some concerns about the long-term sustainability due to the accumulation of herbicide-resistant (HR) weed populations, compacted soil, soil cracks, and varmint (Taxidea taxus) holes causing a rough surface for field operations; the stratification of pH, nutrients, and SOC in the first few cm of the soil profile are additionally problematic [31]. These conditions can lead to a decrease in the plant's ability to supply and absorb nutrients as well as an increased risk of nitrogen (N) and phosphorus (P) losses in surface runoff to the environment [32].

In comparison with conventional tillage, conservation tillage, such as no-tillage or reduced tillage measures, tends to increase the soil's moisture-holding capacity and improve water permeability. Although the soil is susceptible to wind and water erosion [4], the application of minimal tillage systems increases the organic matter content from 0.8% to

22.1% and the stable aggregate content in the water from 1.3% to 13.6%, at 0 to 30 cm deep, compared to the conventional tillage system [33]. This effect could further affect soil water purification and retention functions, particularly under no-tillage conditions [3]. Similarly, two studies [34,35] reported that conservation tillage, such as no tillage and reduced tillage, increases soil organic matter and total soil nitrogen by 18.7% [36,37] and the reverse is true in conventional soil tillage systems (Table 1); (Figures 1 and 2). This indicates that conservation tillage is very important for improving soil productivity, and ultimately, crop yield.

**Table 1.** Effect of soil tillage systems on soil organic matter and soil bulk density.

| Tillage Techniques | 0–8 cm | | 8–30 cm | | 30–40 cm | |
|---|---|---|---|---|---|---|
| | OM | Bd | OM | Bd | OM | Bd |
| **DS** | 1.99 [a] | 1.29 [f] | 1.72 [bcd] | 1.47 [b] | 1.61 [cd] | 1.52 [a] |
| **MT** | 1.82 [b] | 1.33 [e] | 1.54 [d] | 1.45 [bc] | 1.24 [e] | 1.51 [a] |
| **CT** | 1.79 [bc] | 1.39 [d] | 0.76 [f] | 1.41 [cd] | 1.24 [e] | 1.51 [a] |

NB: **DS** = Direct sowing, **MT** = Minimum Tillage, **CT** = Conventional Tillage, **OM** = Organic Matter, **Bd** = Bulk density, letters a, b, c, d, e, f = the significant difference of tillage techniques on organic matter and bulk density [35].

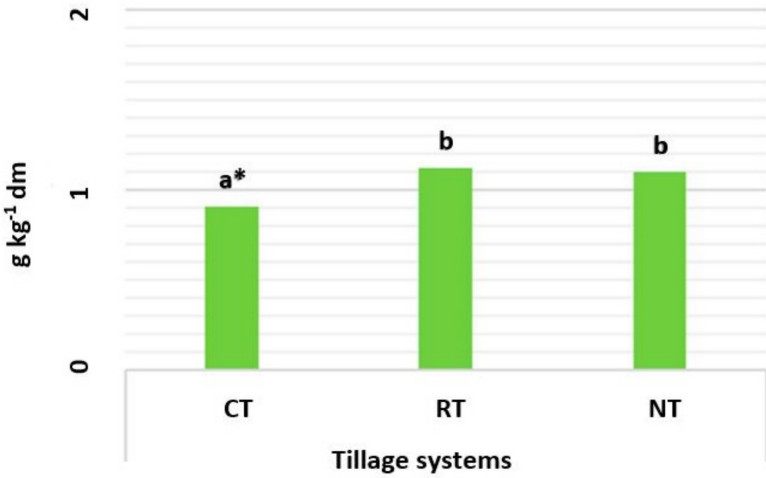

**Figure 1.** The total nitrogen content in the 0–25 cm soil layer, depending on the soil tillage system (average Agriculture of 2014–2016), were CT: Conventional Tillage, RT: Reduced Tillage, NT: Non-Tillage, *: level of significance difference [37].

### 3.2. Effect of Conservation Tillage on Crop Production

Conservation tillage is a farming practice that aims to minimize soil disturbance during planting, thereby reducing erosion and maintaining soil structure and fertility. One of the primary benefits of conservation tillage is the increased crop productivity. By reducing soil disturbance, conservation tillage helps to maintain the structure of the soil, which allows for better water infiltration and retention. This, in turn, leads to improved soil fertility and nutrient availability, which can result in increased crop yields. Numerous studies have examined the effect of conservation tillage on crop productivity and production. For instance, one study [38] found that conservation tillage practices, such as no till and reduced tillage, can increase crop yields by 10–20% compared to conventional tillage methods. Similarly, a meta-analysis [39] revealed that conservation tillage practices increased crop yields by 4.6% on average across different crops and regions. Moreover, a study conducted in the United States found that conservation tillage practices increased corn and soybean yields by up to 3.3% and 0.74%, respectively, compared to conventional tillage methods [40]. However, the effect of conservation tillage on crop productivity and production may vary depending on soil type, climate, crop type, and management practices. For instance,

A meta-analysis of 63 worldwide studies found that conservation tillage increased cereal crop yields but had no discernable effect on legume crop yields [41].

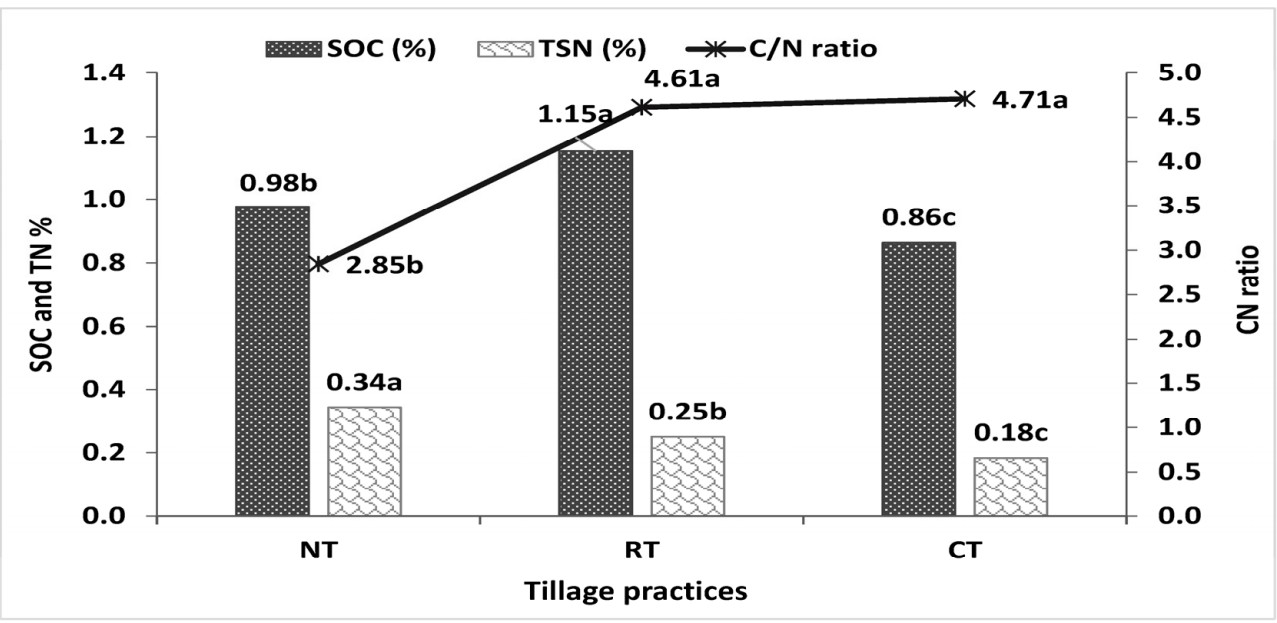

**Figure 2.** The effect of different soil tillage systems on soil organic Carbone (SOC), total soil nitrogen (TSN), and C: N ratio; NT = No tillage, RT = Reduced tillage, and CT = Conventional tillage [36].

In addition to increasing crop productivity, another benefit of conservation tillage is that it can reduce the need for herbicides and other chemicals. By maintaining a permanent soil cover, conservation tillage helps to suppress weed growth, which can reduce the need for herbicides. For instance, a study conducted in Australia found that conservation tillage resulted in lower herbicide use and reduced weed pressure, compared to conventional tillage methods [42].

### 3.3. Effect of Conventional Tillage on Crop Production

Conventional tillage refers to practices considered standard for a specific location and set of crops with the primary intention to bury crop residues. It refers to plowing (inverting the soil) followed by secondary tillage activities, such as tilling and harrowing [43]. This practice is usually considered the basis for determining the cost-effectiveness of erosion control measures [44]. Conventional tillage is used for turning and plowing a deep layer of soil, combining and destroying plant debris, exposing soil pests to the sun to control them, demolition, and ground leveling; this kind of tillage causes excessive soil fragmentation, collapse, and compaction.

On the contrary, conventional tillage also leads to erosion, run-off, impoverishment, and drying out of the land [35]. This could affect soil organic matter and soil bulk density at different soil depths. For instance, as presented in Table 1, the soil organic matter is significantly affected by the conventional tillage system compared to direct seeding and minimum soil tillage systems [34,35]. This implies that a conventional tillage system could affect crop yield and soil productivity, mainly through the decline of soil organic matter status. In addition, under conventional tillage practice, the soil organic carbon and total nitrogen levels were the lowest, leading to the highest C: N ratio (Figure 2).

The conventional soil tillage system affects not only the production and productivity of crop yields but also the total cost of technology implementation during crop implantation [45]. This means that the cost of fuel consumption, the cost of mechanized services, and the time taken to plow the land require maximum costs compared to the conservation soil tillage system.

*3.4. Effect of Soil Tillage System on Soil Carbon Sequestration*

It is internationally accepted that global climate changes result from human intervention in the biogeochemical cycles of water and materials, and soil carbon sequestration practices are considered necessary interventions intended to limit these changes [3]. Soil organic carbon (SOC) plays a significant role in agricultural ecosystems, including soil fertility, soil tilth, nutrient cycling, soil sustainability, and crop productivity, by influencing the physical, chemical, and biological aspects of the soil [46–49]. Soil organic carbon storage in soil-crop ecosystems represents the net balance between the soil carbon breakdown processes brought on by microbial oxidation and the continuous carbon (C) accumulation caused by inputs of crop biomass [49–51]. A positive imbalance results in soil carbon sequestration when carbon dioxide inputs to the soil outweigh carbon dioxide outflow [52–54]. By changing the quantity and quality of crop residues in the soil, the microbial dynamics, and the supply of nutrients, soil and crop management practices (such as tillage, crop type and variety, rotation, chemical fertilizer application, and manure application) could significantly affect the soil organic carbon of cultivated land [46,49,50,55].

Applying agricultural practices to increase organic carbon (SOC) content in the soil is part of a "climate-smart" agricultural approach that sequesters $CO_2$ from the atmosphere [56]. In addition to significantly impacting soil health and agricultural production, the loss of SOC exacerbates climate change. When the soil organic matter (SOM) breaks down, carbon-based greenhouse gases are released into the atmosphere. If this happens at too high a rate, the soil could be contributing to our planet's warming. On the other hand, many soils have the potential to increase their soil organic carbon (SOC) stocks, thereby mitigating climate change by reducing atmospheric $CO_2$ concentrations [57]. The choice of cropping system in farming has a powerful effect on soil health, crop yield, and the broader environment. Conservation tillage with non-inversion techniques will save soil carbon, reduce erosion risk, and enhance soil quality. In addition, it has been demonstrated that conservation tillage sequesters more carbon and total soil nitrogen than conventional tillage (Figure 2).

Stable, well-structured topsoil developed by long-term conservation farming leads to more energy-efficient systems by reducing the energy requirements for farming [58]. Among the significant agricultural strategies, soil organic carbon sequestration is prominent in mitigating greenhouse gas (GHG) emissions, food security enhancement, and agricultural sustainability improvement [59]. According to [12,54] one description, a key carbon sequestration strategy to enhance food security and agricultural sustainability improvement includes (i) restoration of degraded soil through conversion to agricultural land; (ii) application of recommended management practices, such as conservation tillage agriculture, organic farming using manure, and compost; and (iii) using biochar as a soil improver.

## 4. General Overview of Biochar in Agriculture

Biochar is a type of charcoal that is produced through the thermal decomposition of organic material in the absence of oxygen, a process known as pyrolysis [13]. It is also a multifunctional carbon substance, which is used to solve soil fertility and climate change issues [60]. It is considered a novel soil treatment and carbon sequestration pathway that has improved soil structure and ecosystems function due to its complex physical and chemical properties [61]. Due to its importance, the effect of biochar on carbon emissions and sequestration has attracted much attention in recent years. However, additional research on the implications of biochar from an atmospheric-food perspective is required. Biochar has received widespread attention as a viable solution to solving the problems of food security and climate change in agroecosystems, but many questions surround this strategy's potential influence on crop productivity, soil carbon sequestration, and global warming [62]. The application of biochar offers greater advantages regarding soil properties and crop yields in degraded tropical soils compared to those in temperate regions [63,64]. It also improved crop yields more markedly in nutrient-poor and infertile

soils than in healthy and fertile soils [65]. Biochar production can enhance biochar's carbon sequestration potential by up to 45% [66]. Another study described that biochar has the potential to capture and store between 0.6 and 11.9 gigatons of carbon dioxide annually, with the primary determining factor being the accessibility of biomass resources for the production of biochar [67] (Figure 3). Additionally, studies have found that the potential for GHG mitigation from biochar increases over time. Specifically, the studies estimate that by 2030, biochar could mitigate between 1 and 1.8 billion metric tons of $CO_2$ per year [14,68–70]. By 2050, this potential increases to between 1.8 and 4.8 billion metric tons of $CO_2$ per year [71–73], and by 2100, the range is estimated to be between 2.6 and 4.8 billion metric tons of $CO_2$ per year [74]. These estimates suggest that biochar has the potential to be an important tool for mitigating GHG emissions and addressing climate change, especially as its use is scaled up over time. Due to its high stability and large-scale production potential, the application of biochar has a high potential for long-term SOC sequestration and is limited only by the available biomass [75–79].

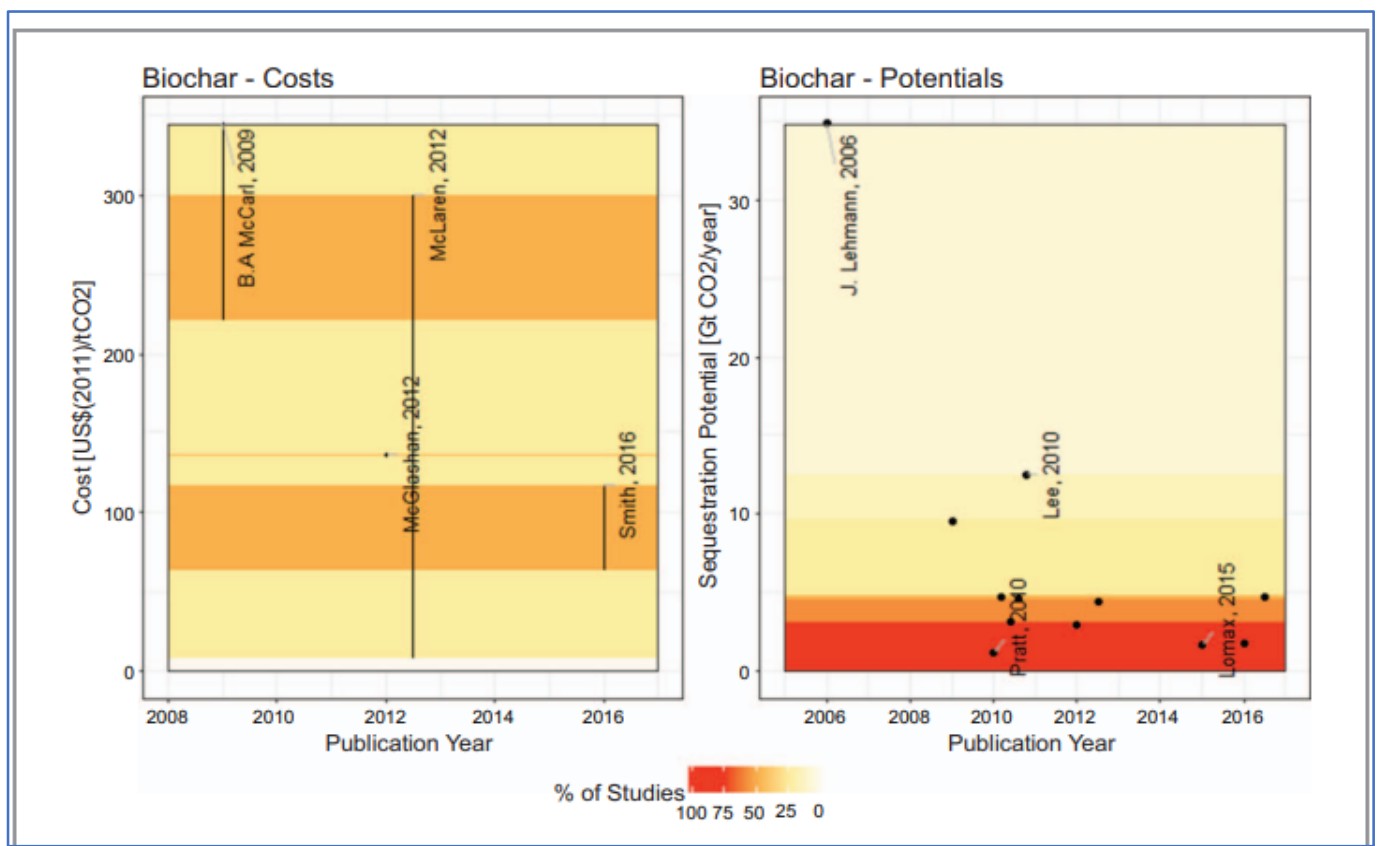

**Figure 3.** Costs and carbon sequestration potentials for biochar. Estimates and ranges at the top and bottom end of the distribution are labeled; the data can be further explored in our online supporting material available at https://mcc-apsis.github.io/NETs-review/ accessed on 22 May 2018 [67].

*4.1. Making, Production, Processing, and Digestion of Biochar*

The production, processing, making, and digestion of biochar offer a wide range of opportunities for sustainable agriculture, waste management, and energy production. However, it is important to carefully consider the feedstocks, processing methods, and application rates of biochar to ensure its effectiveness and minimize any negative impacts on the environment (Table 2).

**Table 2.** Short Summary of Making, Production, Processing, and Digestion of Biochar.

| Process | Description | Reference |
|---|---|---|
| Making | Biochar can be made from a wide range of feedstocks, including agricultural residues, forestry wastes, and even municipal solid waste. The choice of feedstock and pyrolysis conditions affects the properties of the biochar and its suitability for different applications. For example, biochar made from hardwoods may have a higher carbon content and be more stable than biochar made from softwoods. | [80,81] |
| Production | Biochar is produced through the process of pyrolysis, which involves heating biomass in the absence of oxygen. This process converts the biomass into a carbon-rich material that is resistant to decomposition. The temperature of pyrolysis affects the properties of the biochar, such as its porosity and surface area. | [13] |
| Processing | After biochar is produced, it can be processed to improve its properties for specific applications. This may involve crushing, sieving, or adding amendments to the biochar. For example, biochar can be impregnated with nutrients to make it more effective as a soil amendment. | [82] |
| Digestion | Biochar can be used as a feedstock for anaerobic digestion, a process that converts organic matter into biogas. When biochar is added to an anaerobic digester, it can improve the performance of the system by providing a surface for bacterial growth and removing inhibitory substances from the digester. | [83] |

### 4.2. Effect of Biochar on Soil Properties

Biochar is a type of charcoal that is created by heating organic materials, such as wood, agricultural waste, and other types of biomasses in the absence of oxygen. The resulting material is rich in carbon and is used to improve soil fertility, water retention, and carbon sequestration. Biochar has been found to improve soil structure by increasing the amount of pore space and reducing soil compaction. This allows water to penetrate deeper into the soil, reducing runoff and increasing water retention. In one study [84], the addition of biochar to soil increased the water-holding capacity by up to 18% compared to control soils. It has been also used to increase the nutrient use efficiency of plants. For instance, another study [85] revealed that biochar increased the utilization efficiency of nitrogen (NUF), phosphorus (PUE), and potassium (KUE) in quinoa plants and could be utilized as an organic fertilizer for crops [86]. Additionally, according to another study [87], biochar application has a positive effect on soil properties, as evidenced by an increase of 1.5 times Mn in soil and 1.4 times total organic carbon, as well as the neutrality observed in soil pH and the availability of P and cations, total N, and other extractable soil nutrients other than Mn during a two-year trial in Idaho, United States. In connection with this, an optimistic effect was detected when biochar was applied throughout three consecutive years of field trials: a minor increase in soil pH and increased potassium nutrient levels [88]. Moreover, a positive effect was also observed on soil pH, P, available SOC, hydraulic conductivity, water-holding capacity, N content, $NH_4 + $-N and $NO_3$-N in soil, soil microbial enhancement, and enzyme activity through the use of biochar during a 5-year field trial in the subtropical region of Jianxi, China [89]. These studies suggest that a biochar amendment is a practical approach to improving soil structure, carbon content, soil availability, the efficiency of nutrient use, and the influence on soil microbes.

The addition of biochar to soil has been shown to increase soil carbon sequestration, which can help mitigate climate change. In a previous study [90], the addition of biochar to soil increased carbon sequestration by up to 0.4 t C/ha/yr. Similarly, biochar significantly improves soil quality, carbon sequestration, and greenhouse gas ($CO_2$, $N_2O$, $CH_4$) emission reduction [91,92]. In addition, other study demonstrated that applying biochar might decrease soil nitrous oxide ($N_2O$) emissions by 28%, with the total being depleted to 60% [62,93–97]. Moreover, Biochar has been found to enhance soil microbial activity by providing a habitat for beneficial microorganisms and improving soil nutrient availability. According to one study [98], the addition of biochar to soil increased the activity of soil microorganisms by up to 46%. Furthermore, it has unique adsorption and desorption mechanisms that control nutrient leaching [99] and can enhance plant productivity, particularly

when coupled with other organic products, such as manure and compost; sewage sludge can be used as a soil amendment [100]. In contrast, there has been a lot of diversity and confusion in the results about the impact of biochar on crop output, soil organic carbon (SOC), and greenhouse gas emissions [101]. This suggests that further research is needed with other agronomic practices, such as soil tillage, to fully understand the effects of biochar on agricultural systems.

The amount of biochar required to increase soil organic carbon stock depends on various factors, including the frequency of application. Studies have shown that repeated applications of biochar increase soil organic carbon stock more than a single application. For example, some studies have found that one application of biochar can increase the soil organic carbon pool by 26%, while consecutive applications can increase it by an average of 55% [79,102]. Moreover, different types of biochar have different effects on soil properties [101]. The structural components of the raw materials used to produce biochar can affect its ability to modify soil properties [103]. Therefore, it is essential to consider the characteristics of the biochar before applying it to the soil.

*4.3. Effect of Biochar on Crop Production and Crop Quality*

Biochar has been shown to have beneficial effects on crop production and crop quality. It has a high surface area and a porous structure, which allows it to absorb and retain water and nutrients, and to provide a habitat for beneficial microorganisms. This can lead to increased crop yields and improved crop quality. Several studies have reported the positive effects of biochar on crop production. For instance, research conducted by one researcher [94] found that improved crop productivity raised yields by an average of 11.0%. Another study [104] found that the addition of biochar to unfertilized soils at rates of 10 t ha$^{-1}$ and 40 t ha$^{-1}$ increased rice yields by 12% and 14%, respectively; rice yields increased by 8.8% and 12.1% at the same rates of 10 t ha$^{-1}$ ha and 40 t ha$^{-1}$ in soils with N fertilization, respectively. Similarly, it can increase plant productivity by an average of 10%, due to the increased water-holding capacity and nutrient availability of the soil, but the impact on yield can vary depending on factors such as soil type, environment, and management conditions [61,90]. Furthermore, a study conducted in Brazil found that biochar application increased soil pH and improved nutrient availability, resulting in increased sugarcane yield and quality [105]. It can also alter the levels of available nutrients in the soil, which can increase the plant's ability to absorb carbon dioxide from the atmosphere [106]. In addition to its effects on soil and nutrient availability, biochar can also have direct effects on plant physiology and biochemistry, leading to improvements in plant growth and health [107].

Biochar has also been shown to improve crop quality. For example, a study conducted in Malaysia discovered that combining biochar with other soilless media boosted growth and yield without harming the post-harvest quality of lowland cherry tomatoes [108]. In addition to its effects on soil fertility and crop quality, biochar has also improved drought tolerance in maize plants [109]. Overall, the use of biochar has shown promise as a tool for improving crop production, crop quality, and disease suppression, while also potentially contributing to climate change mitigation efforts.

*4.4. Effect of Biochar on the Environment*

Biochar has been touted as a potential tool for mitigating climate change by sequestering carbon in the soil, improving soil fertility, and reducing greenhouse gas emissions from agricultural practices (Table 3). However, the environmental impact of biochar production and use is not fully understood and requires further research. According to some researchers [110,111], biochar may naturally contain pollutants that were either added by the feedstock (such as heavy metals) or co-produced during (improper) pyrolysis (such as polycyclic aromatic hydrocarbons). This leads to increased soil acidity and the leaching of heavy metals into the environment. Similarly, another study [13] found that biochar produced from feedstocks with high concentrations of heavy metals, such as chicken manure,

could potentially lead to increased concentrations of heavy metals in the soil and water. In general, further research is needed to fully understand the environmental implications of biochar production and use.

**Table 3.** Short Summary of General Effect of Biochar on Soil, Yield, and Environment.

| Biochar Effect | Reference |
| --- | --- |
| Improves soil fertility | [98,112] |
| Increases crop yields | [82,90] |
| Reduces greenhouse gas emissions | [74,81] |
| Improves water retention | [113,114] |
| Reduces soil erosion | [113,114] |
| Enhances nutrient cycling | [81,98] |
| Suppresses soil-borne pathogens | [90,98] |
| Improves soil structure | [112,114] |
| Reduces leaching of nutrients | [105,115] |
| Reduces leaching of pollutants | [116] |
| Improves plant growth and health | [117,118] |
| Increases carbon sequestration | [13,74] |

*4.5. Biochar Dose Optimization for Crop Yield and Cost*

Biochar is a valuable soil amendment that can improve soil fertility, increase its water-holding capacity, and reduce greenhouse gas emissions. However, the optimal dose of biochar required for achieving maximum yield and cost-effectiveness is unclear and depends on various factors, such as soil type, crop type, and biochar properties. Therefore, building an optimization model can help determine the optimal dose of biochar required for maximum yield and cost-effectiveness. Several studies have explored the relationship between biochar dosage, yield, and cost indicators. For example, some researchers [119] have determined the optimal biochar dose for sugar beet yield. They found and recommend that the maximum yield was achieved at a biochar dose of 10 t ha$^{-1}$ year$^{-1}$ with minimum production cost. Additionally, another researcher [120] found the dose of biochar needed for the highest productivity and cost-effectiveness in maize yield. The study discovered that adding 1% biochar resulted in a 22% increase in maize yield, which could generate EUR 620 per hectare in revenue on a maize purchase price of EUR 117 per ton. However, other factors, such as fluctuating costs, government subsidies, environmental restrictions, and market demand, could also affect the economics of biochar production and application. Moreover, a three-year agronomic trial was conducted with maize and mustard farming to study the economic viability of biochar addition. The study found that the optimal dose was determined to be 15 t ha$^{-1}$ based on several considerations, including agronomical factors, such as crop yield; economic factors, such as cost-benefit analysis; and environmental factors, such as carbon sequestration [121].

**5. Interactive Effects of Tillage System and Biochar Application on Soil and Crop Productivity**

Organic nutrient amendments and tillage systems have been frequently suggested as critical aspects of food security under the changing climate, however, responsible agronomic-based solutions should follow a food system and circular economy approaches. The adaptation of minimum tillage in combination with organic amendments, including biochar, may sustain the soil health indicators i.e., increase nitrogen status, soil organic carbon, and phosphorus [122] to support crop growth and environmental sustainability by reducing soil erosion and carbon emissions [123]. It is also worth mentioning that a combined effect of biochar application and deep tillage has significantly decreased bulk density, increased the air capacity of the soil, and enhanced the soil organic carbon content and available phosphorus, which subsequently enhances the grain yield of crops [124,125]. Higher yield, soil organic matter, soil available nitrogen, phosphorus, and potassium have

been also observed under conservation tillage compared with the conventional tillage system [126].

However, the extent and magnitude of the combined effect of tillage and biochar application on crop yield and soil health indicators was found to be inconsistent. For instance, the application of biochar had a more profound effect on crop yield and the productivity of the soil under reduced tillage than under other tillage operations [127]. Higher soil productivity under deep tillage could be due to an invert and a complete mixture of the soil, which further turned the top soil layer into the deeper soil profile, and facilitated the decomposition rate of biochar [128]. These techniques directly influence soil microbes by adding nutrients and indirectly influence them by changing the soil characteristics [129,130]. On the other hand, biochar and conservation farming (CF), which incorporates basin tillage, residue retention, and crop rotation, may assist in reducing the detrimental effects of conventional agriculture [131]. This implies that there is a symbiotic relationship between tillage practices and an exogenous application of biochar, although the magnitude and level of influence of tillage practices rely on the soil and environmental conditions.

## 6. Conclusions

Agriculture is the backbone of the worldwide economy as it supports different crop yields and animal products. To obtain the required amount and quality of crops, the soil health and the environment should be sustained by undertaking different activities. Among those actions, appropriate agronomic practices, such as proper soil tillage systems and multifunctional use of nutrient application systems, have a significant role in increasing the quality and productivity of the crop yield, improving soil health, and increasing environmental protection as well. Specifically, soil tillage is one of the critical components of agricultural systems and practices, commonly working internationally in croplands to reduce climatic and soil restrictions, even though sustaining several ecosystem services is an issue. It is a crop production factor contributing up to 20% and affecting the sustainable use of soil resources through its influence on the physical and chemical properties of soil. The system of tillage is categorized into conservation tillage or conventional tillage systems. Conservation tillage attains the highest organic matter accumulation, achieves the maximum root mass density (0–15 cm soil depth), improves physical and chemical properties of the soil, sequesters more carbon, and can increase crop yields by 4.6% on average across different crops and regions compared to conventional tillage systems. In addition, biochar is a multifunctional carbon substance, which is used to solve soil fertility and climate change issues. It is considered a novel soil treatment and carbon sequestration pathway that has improved soil structure and ecosystems function due to its complex physical and chemical properties. The use of biochar can enhance soil fertility, reduce soil erosion, decrease soil nitrous oxide ($N_2O$) emissions by 28% (down to 60%), promote microbial activity, increase carbon sequestration, and increase crop yields by 11% on average.

Therefore, it is clear that the soil tillage system and biochar application have a symbiotic and asymmetric relationship that can promote soil carbon sequestration, improve soil properties, and improve crop yields. The interactive effect of soil tillage and biochar showed a positive effect on crop yield and soil properties in conservation and conventional tillage systems. However, the extent and magnitude of the combined effect of tillage and biochar application on crop yield and soil health indicators was found to be inconsistent. This means it is difficult to detect the most advantageous soil tillage system with the combination of biochar to improve soil health indicators, environmental protection, and crop yield. Thus, to detect the most appropriate soil tillage system with the combination of biochar, further research will be needed in different environments, soil types, seasons, and crop types.

**Author Contributions:** Conceptualization, A.A.B., A.A.M. and A.P.; Methodology, A.A.B.; Writing Original draft preparation, A.A.B., A.A.M. and A.P.; Writing review and editing, A.A.B., A.A.M. and A.P. All authors have read and agreed to the published version of the manuscript.

**Funding:** This work was supported by the Tempus Public Foundation (Hungary) within the framework of the Stipendium Hungaricum Scholarship Programme, Hungarian University of Agriculture and life Sciences (MATE), supported by Ministry of Education of Ethiopia and Mekdela Amba University.

**Institutional Review Board Statement:** Not applicable.

**Informed Consent Statement:** Not applicable.

**Data Availability Statement:** Data included in article referenced in article.

**Acknowledgments:** This review paper is benefited a lot from the work of other scholars working around the subject matter and hence, we give our sincere thanks though we have provided full reference to the information used, we give our thanks also Tempus Public Foundation (Hungary) for providing the opportunity.

**Conflicts of Interest:** The authors declare no conflict of interest.

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
