# Peer review of "Symbiotic and Asymmetric Causality of the Soil Tillage System and Biochar Application on Soil Carbon Sequestration and Crop Production"

_soilsystems, doi:10.3390/soilsystems7020048_

Round 1
Reviewer 1 Report
1. The work was done on a topical issue.
2. There are no requirements for the content of various crops in soils.
3. The processes are not fully described: production; processing; making; digestion of biochar. It was necessary to consider the timing and linkage with the achievement of targets.
4. Perhaps it was worth building an optimization model for the relationship between the dose of biochar, yield and cost indicators.
Author Response
Dear Reviewer, Thank you so much for your valuable comments and suggestions. We tried to address the comments in the manuscript. please see the attachment

Reviewer 2 Report
Manuscript ID: soilsystems-2313784
The manuscript “Symbiotic and Asymmetric Causality of the Soil Tillage System and Biochar Application on Soil Carbon Sequestration and Crop Production”, which is authored by Bogale et al. Therefore, the manuscript is well-written and adds some new insights into the topic. Therefore, the manuscript will be of interest to its readers. However, to further improve its quality, I have suggested some modifications in the manuscript as below:
1. There are several English language and grammatical mistakes in the manuscript, which must be corrected before considering this manuscript for publication.
2. Improve the quality of figures 1 and 3
3. Statistical design and analysis not clear to me, so it needs to be clarified.
4. What does figure 2 do?
5. Please clearly mention the hypothesis of this study.
6. Please improve conclusion section.
7. Double check the citations and their references in the list.
Author Response
Dear Reviewer, thank you so much for your valuable suggestions and comments. We tried to incorporate the comments in the manuscript. Please see the attachment of the reviewer's point-by-point responses.

Reviewer 3 Report
Dear Authors:
The study “Symbiotic and Asymmetric Causality of The Soil Tillage System and Biochar Application on Soil Carbon Sequestration and Crop Production” highlighted the effect of conservation tillage and biochar application on productivity, soil changes and environmental effects. The paper is written well, explained comprehensively and will be of interest to the readers of the Journal.
General comments are:
1. Section 3.1. Effect of conservation tillage on crop production–the section specified the tillage effect on productivity but very studied reviewed on the aspect while most of the cited studies on effect on soil. So, please add few more study with their place to pin picture the effect on crop yield.
Moreover, thew effect on soil and crop productivity may be divided in separate sub-sections.
2. The review title highlighted the effect of biochar specifically. Accordingly, the section 4 needs more elaborative discussion on effect of biochar on soil, crop and environment, separately through sub-sections of section 4.
The overall effect of biochar may also be summarized in table for more visibility and comparison.
A graphical representation of biochar effects on soil, crop and environment may also add value to paper.
3. Lines 289-292: Recent assessments on the potential and cost of biochar estimate that biochar could sequester between 0.6 Gt CO2 yr−1 and 11.9 Gt CO2 yr−1, mainly depending on the availability of biomass for biochar production (Figure 3). Not clear….please explain clearly.
4. Lines 318-319: Hence, application of biochar combined with deep tillage practice could be a promising agronomic practice to overcome problems associated with shallow ploughing layers and poor release of nutrients, particularly under clay vertisols [98].
Both conservation tillage and deep tillage provided benefits in 1-2 studied (Line 312-317). So, how it can be concluded that biochar benefits more under deep tillage.
More studied should be presented and conclusion may be revised accordingly, taking into consideration the soil texture, if any contrast effect was there.
5. The sentences of conclusion (A combined use of deep tillage……………………enhances the crop yield. Hence,…………. dependency of chemical fertilizers) are contradictory to each other. Please revise in light of comment3.
Thanks
Author Response
Dear Reviewer, we appreciate your concrete and valuable comments and suggestions. We tried to incorporate the comments in the manuscript. please see the attachment.

Reviewer 4 Report
This paper does provide a review of tillage-biochar contributions to regenerative agriculture.
I do not understand what the title means or is attempting to say. The current title is what some would call "obfuscation".
I personally believe that the way that biochar is created (heating procedure) affects its chemical properties, particularly with its nutrient exchange capabilities, and this statement is not in the paper. I've added a reference below that the authors may want to use to clarify this point.,
specific comments:
Abstract. Add std errors or some estimate of uncertainty to numbers reported.
Line 43 ‘drivers’
Paragraph 49: need subscripts on molecules
Line 68 “stable” rather than steady?
Line 86: selected articles were
Line 104: ‘this’ not these.
Line 117 ‘reduced’ subsurface…
Line 137 “provides all of the aforementioned…
Figure 1. we need some estimate of variability on these estimates.
Table 1. Where are these data from? Readers should not have to go to the references to find this.
Figure 2. location of study?
line 209... Your focus on tillage is not universally accepted. See citation below.
Line 233-234 Not clear what is means.
Lines 276-77 clarify what a biochar ratio is.
Line 322. “was found to be inconsistent.”
Line 336-338 Sentence not clear.
Reference #4 is not correct. I did not check others.
Another fact about non-inversion conservation tillage?
Öttl, L.K., Wilken, F., Hupfer, A. et al. Non-inversion conservation tillage as an underestimated driver of tillage erosion. Sci Rep 12, 20704 (2022). https://doi.org/10.1038/s41598-022-24749-7
The main finding of the present work revealed that the high pyrolytic temperatures in BC production may have negative impacts on phyto-availability of essential nutrients. Depending on the feedstock raw material and pyrolysis process used for producing BC, it has different capacities for releasing nutrients in the soil. An economically feasible method of producing newly engineered biochar, with more controlled pyrolysis and C-based materials, for suitable agriculture needs to be developed.
Elkhlifi, Z., Iftikhar, J., Sarraf, M., Ali, B., Saleem, M.H., Ibranshahib, I., Bispo, M.D., Meili, L., Ercisli, S., Torun Kayabasi, E. and Alemzadeh Ansari, N., 2023. Potential role of biochar on capturing soil nutrients, carbon sequestration and managing environmental challenges: a review. Sustainability, 15(3), p.2527.
Author Response
Dear Reviewer, Thank you so much for your important comment and suggestions. We tried to incorporate the comments in the manuscript. Please see the attachment.

Round 2
Reviewer 1 Report
1. It is necessary to specify the conclusions based on the work done.
2. It is necessary to enrich the conclusions with the numerical material presented in the paper.
Author Response
Dear reviewer, thank you so much for your valuable comments. We have incorporated the comments in the manuscript. please see the attachment
